# Effect of Dispersion by Three-Roll Milling on Electrical Properties and Filler Length of Carbon Nanotube Composites

**DOI:** 10.3390/ma12233823

**Published:** 2019-11-21

**Authors:** Ji-Hwan Ha, Sang-Eui Lee, Sung-Hoon Park

**Affiliations:** 1Department of Mechanical Engineering, Soongsil University, 369 Sangdo-ro, Dongjak-gu, Seoul 06978, Korea; jhwan618@gmail.com; 2Department of Mechanical Engineering, Inha University, Inha-ro 100, Michuhol-gu, Incheon 22212, Korea

**Keywords:** carbon nanotube, composite, three-roll milling, CNT dispersion, filler length variation

## Abstract

For practical use of carbon nanotube (CNT) composites, especially in electronic applications, uniform dispersion of a high concentration of CNTs in a polymer matrix is a critical challenge. Three-roll milling is one of most reliable dispersion techniques. We investigate the effect of three-roll milling time on CNT length and the electrical properties of a CNT/polydimethylsiloxane composite film with 10 wt% CNTs. During the milling process, the CNT length is decreased from 10 to 1–4 μm by mechanical shear forces. The electrical conductivity increases after 1.5 min of milling owing to dispersion of the CNTs but decreases with increasing milling time owing to the decrease in the CNT length. Considering the changes in the electrical conductivity of the CNT composite and CNT length, we determined how to optimize the three-roll milling time to obtain a suitable dispersion state.

## 1. Introduction

Carbon-based nanomaterials are used to improve the electrical conductivity of composites owing to their superior electrical properties [1,2]. Many carbon-based nanomaterials are used as conductive fillers, for example, carbon black, carbon nanotubes (CNTs), and graphene. CNTs have been investigated for use in various applications owing to their superior electrical, thermal, and mechanical properties [3,4,5]. In addition, CNTs have a large aspect ratio and are chemically stable. Consequently, CNTs have been mixed with a polymer to fabricate conducting composites [6,7,8]. Tube-shaped conductive fillers can easily form connected networks in a polymer. The conductive pathway in the composite improve its electrical properties [9]. Furthermore, the aspect ratio of CNTs is related to their electrical properties. According to the literature, the electrical conductivity is enhanced by increasing the aspect ratio of the filler because the resulting electrical network includes many contact points between conductive fillers [10,11]. In addition, long CNTs can easily form a dense electrical network. Therefore, longer filler length could improve the electrical performance of composites.

To maximize the electrical and mechanical properties of composites, CNTs must be dispersed uniformly in the polymer matrix. However, nano-size carbon fillers such as CNTs, graphene, and carbon black often appear as large bundles or aggregates owing to the strong van der Waals forces, which produce a non-uniform dispersion [12,13,14,15]. Sonication is typically used as a traditional lab-scale method for dispersion of CNTs [16,17,18,19,20], in particular to disperse aggregated CNTs in solution. However, this method cannot be used when the filler concentration is high owing to the high viscosity of the CNT solution. In addition, over time, ultrasonically dispersed CNTs in solution re-aggregate owing to the low viscosity of the CNT solution. To address these limitations, many researchers have developed alternative dispersion methods. Three-roll milling is an effective method for dispersing a high filler content to improve many composites [4,21,22,23,24,25,26]. This method uses the mechanical shear forces by decreasing the gap between the rolls. The CNTs in the mixing paste are uniformly dispersed by passing through the microscale gap [22,23]. Furthermore, the number of three-roll mill passes (milling time) affects the CNT aspect ratio.

In this study, we fabricated CNT/polydimethylsiloxane (PDMS) composite films with a high CNT content (10 wt%) to obtain an electrically conductive composite. We used three-roll milling to obtain a uniform dispersion in the CNT composite. The effect of the milling time on the filler length was observed by scanning electric microscopy (SEM). In addition to the morphological analysis, Raman spectroscopy was used to determine the changes in the carbon structure of the CNTs. Furthermore, we measured the electrical properties associated with the length variation resulting from differences in the three-roll milling time.

## 2. Materials and Methods 

Multi-walled CNTs (CM250, 15 nm in diameter and 10–15 μm in length, Hanwa Nanotech, Seoul, South Korea) were used as a conducting filler. PDMS was used as a polymer matrix (Dow Corning, Midland, MI, USA, Sylgard 184). A paste mixer (Daehwa, Seoul, Korea) and three-roll mill (Intech, Gyeonggi-do, Korea) were used to obtain a high-viscosity CNT paste. The CNT contents of the composite paste is 10 wt%. First, CNT and PDMS (elastomer/curing agent = 10:1) were premixed by a paste mixer at 500 rpm for 30 s and 1500 rpm for 60 s. Next, three-roll milling for various durations was conducted to disperse the CNTs. Seven composite pastes were prepared by milling for 0, 0.5, 1.5, 2, 3, and 6 min. The gaps between the rolls are 5 μm (front gap) and 12.5 μm (back gap) each. For electrical measurement of the composite, CNT composite films were fabricated by hot film pressing (Qmesys, Inc., Gyeonggi-do, Korea) at 150 °C for 40 min. 

The CNT length after three-roll milling was observed using SEM (XL30, Phillips, North Billerica, MA, USA). Chloroform (solvent) was used to dissolve the PDMS in the CNT paste. Sonication was conducted at a frequency of 20 kHz for 20 min. After the sonication process (Branson 450, VWR, Radnor, PA, USA), the CNT solution was spin-coated on a Si wafer. The spin coating conditions were 500, 1500, and 500 rpm for 30, 60, and 30 s, respectively. The coated wafer was dried at room temperature overnight. The wafers that is coated the composite pastes were observed in 5 kV condition without conductive costing by SEM. For the measurement of CNTs length on the composite, we observed SEM images. Furthermore, we calculated the distributed length of CNTs according to three-roll milling time.

After CNT/PDMS film fabrication, UV etching was conducted for 300 s to ensure electrical contact by UV ozone cleaner (Jaesung engineering Co., Seoul, Korea). Silver paste (Protavic, Levallois-Perret, France) was used as an electrode; it was coated on the film and cured in an oven at 175 °C for 1 h. Next, a four-wire resistance method (Keithley 487 picoammeter and Keithley 2400 sourcemeter, Keithley, Cleveland, OH, USA) was used to measure the resistance of the composites. 

## 3. Results and Discussion

The dispersion of CNTs in the polymer was performed by applying mechanical shear forces in three-roll milling. Figure 1 schematically illustrates the dispersion mechanism of the three-roll mill. Shear forces are applied by the microscale gaps between the rollers. CNTs are bundled by the van der Waals force before milling. However, the shear forces of the three-roll mill can produce a uniform dispersion of the fillers. Therefore, this method is typically used to disperse a carbon-based filler in a composite. CNT aggregation negatively affects the electrical, thermal, and mechanical properties of the composite. For example, agglomeration of CNTs in a composite can decrease the electrical conductivity because there are fewer contacts between CNTs in the electrical pathway of the composite. Therefore, a uniform dispersion is needed to improve the electrical properties. However, three-roll milling can have negative effects. Although it can disperse the conductive filler in the composite, increasing the roll milling time decreases the CNT aspect ratio. Specifically, when three-roll milling is performed for longer than 1.5 min, the CNTs become shorter as a result of high shear forces by the microgap in the three-roll mill. Therefore, the duration of dispersion by milling must be optimized to improve the electrical properties.

Figure 2 shows SEM images of CNTs in the CNT/PDMS paste. With no milling, the fillers are approximately 9–10 μm long. After 1.5 min of milling, the filler size decreases remarkably, as shown in Figure 2b. CNTs with lengths of 4–5 μm are dispersed on the wafer. When the dispersion time increases further to 3 min, the CNTs are even shorter (2.5–3 μm). Furthermore, the length of the filler is as short as 1–1.5 μm after 6 min of milling time. The length decreases with increasing dispersion time because three-roll milling applies pressure and shear forces to the CNTs.

To confirm the characteristics of the CNT composite, Raman spectra are obtained. Figure 3 shows the Raman spectra of pristine CNT and composites (6 min). The D band (representing defects and disorder) and G band (representing carbon structure) in the spectra are identical. In addition, the I_D_/I_G_ ratios of pristine CNT and CNT/PDMS composite are 1.37 and 1.31, respectively. Therefore, the characteristics of the carbon atoms in the CNT filler are not changed (such as atomic destruction) by three-roll milling. Furthermore, the changes in filler length during three-roll milling do not affect the nature of the CNTs.

Figure 4 shows the length distributions of CNTs milled for various durations. Without three-roll milling, the length of the CNTs is distributed almost entirely between 7 and 14 μm, and most of the CNTs are 10 μm long (this is the original length). However, the CNT size can be controlled by using three-roll milling. After milling for 1.5 min, the length decreases from 10 to 4 μm, and the length distribution ranges from 1 to 8 μm owing to the shear forces applied by the three-roll mill. The length is decreased to 0.5–3.5 μm by increasing the milling time to 3 min, and most of the CNTs are shorter (3 μm) than those obtained by 1.5 min of milling. The length of the CNTs milled for 6 min has the same range as that of the CNTs milled for 3 min (Figure 4d). However, most of the CNTs milled for 6 min have a length of 1 μm. Therefore, increasing the milling time decreases the CNT aspect ratio because more force is applied to the carbon fillers during a longer milling time.

Figure 5 shows SEM images that reveal the morphology (i.e., dispersion state) of CNT/PDMS films fabricated using three-roll milling for 1, 1.5, and 6 min. For 1 min of milling, the degree of dispersion is poor owing to the short milling time. For 1.5 min of milling, the CNTs are dispersed well owing to the moderate milling time. When the milling time is increased to 6 min, the composite morphology shows the most uniform dispersion. Therefore, a long three-roll milling time results in a uniform dispersion and short CNTs. Thus, when three-roll milling is used, the dispersion state and the CNT length have a trade-off relationship. 

We measured the electrical properties of composites obtained using various dispersion times. Figure 6a shows that the CNT length is decreased by increasing the milling time because of the mechanical shear forces. The error bar shows all CNT length distribution. As the milling time increases, the length of CNTs becomes more uniform. The length with no dispersion is distributed between 7 and 14 μm. Figure 6b shows the conductivity of CNT/PDMS films (CNT content: 10 wt%). The maximum conductivity of 372 S/m is measured after 1.5 min of milling time. Furthermore, this initial increase in the conductivity indicates that the composite contains many contacts owing to the large aspect ratio of the CNTs. For milling times longer than 1.5 min, the conductivity decreases because the aspect ratio of the fillers decreases. For milling durations of 2, 3, and 6 min, the electrical conductivity is 297, 280, and 248 S/m respectively. The number of contacts between fillers decreases because the short CNTs are not connected to each other. On the other hand, the non-uniform dispersion case (1 min milling) results in large variations in conductivity and relatively lower conductivity. With increasing milling time, the dispersion state becomes more stable, and CNT length decreases. The more stable dispersion state means homogenous dispersion of CNTs without aggregation of fillers. At this state, the aggregation of CNTs is not observed through SEM imaging. As a result, although the conductivity decreases, it varies less across the film. This result indicates that the aspect ratio of the filler and the dispersion state have a trade-off relationship during three-roll milling. Therefore, in terms of high conductivity with moderate error bar, 1.5 min milling time is optimal condition for the fabrication of the CNT/PDMS composites. In addition, the electrical conductivity and its variation can be easily tuned by controlling the three-roll milling time.

## 4. Conclusions

We fabricated composite films with a high CNT content (10 wt%) to observe the effect of three-roll milling time on filler length and film electrical properties. The filler length and dispersion state are related to three-roll milling time. Dispersion times of 0, 1, 1.5, 3, and 6 min were applied. SEM images showed that as the milling time increased, the CNT length decreased, which was attributed to the application of mechanical forces. In addition, electrical conductivity was highest when the milling time was 1.5 min because a uniform dispersion of CNTs and a suitable filler length were obtained. However, the electrical conductivity decreased sharply when the milling time exceeded 1.5 min because the CNTs became shorter. There is optimal milling time for high conductivity and small conductivity deviation, which enable mass production of CNT composites. 

## Figures and Tables

**Figure 1 materials-12-03823-f001:**
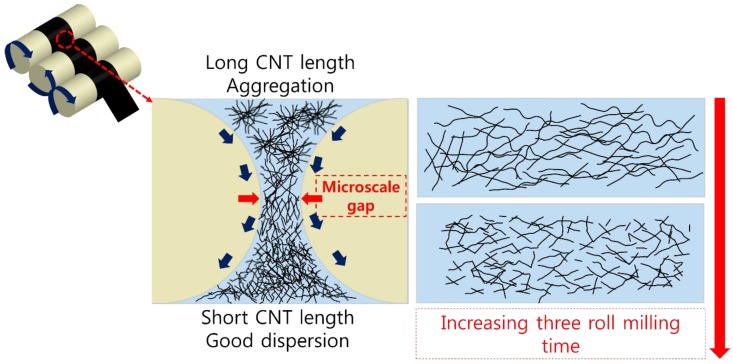
Mechanism of CNT (carbon nanotube) dispersion and aspect ratio changes by shear forces during three-roll milling. Filler length is reduced by increasing the three-roll milling time.

**Figure 2 materials-12-03823-f002:**
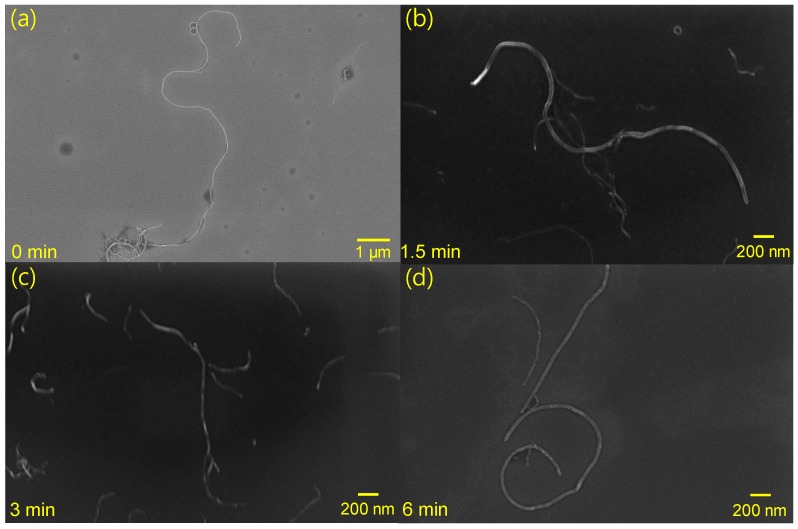
SEM images showing length of CNTs in CNT/PDMS paste after three-roll milling for (**a**) 0 min, (**b**) 1.5 min, (**c**) 3 min, and (**d**) 6 min.

**Figure 3 materials-12-03823-f003:**
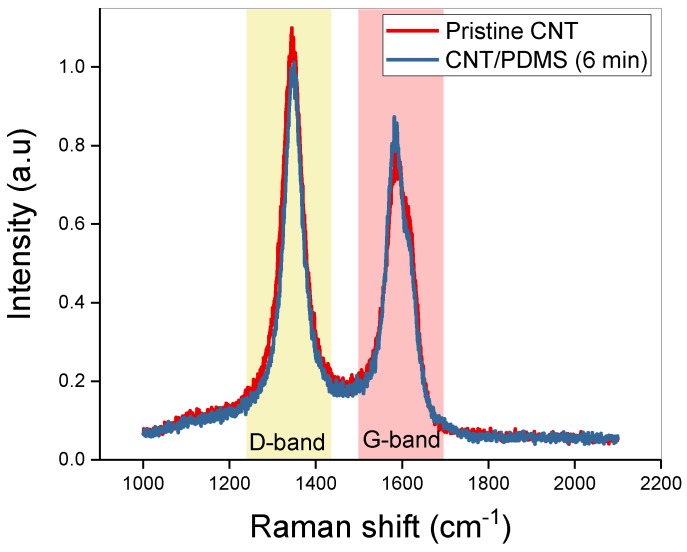
Raman spectra of pristine CNT and CNT composites.

**Figure 4 materials-12-03823-f004:**
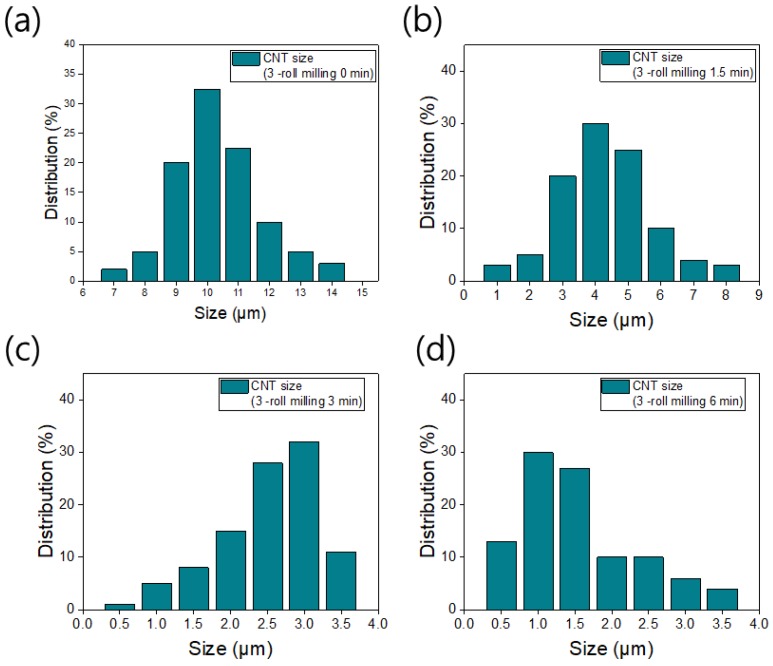
Length distributions of CNTs milled for (**a**) 0 min, (**b**) 1.5 min, (**c**) 3 min, and (**d**) 6 min.

**Figure 5 materials-12-03823-f005:**
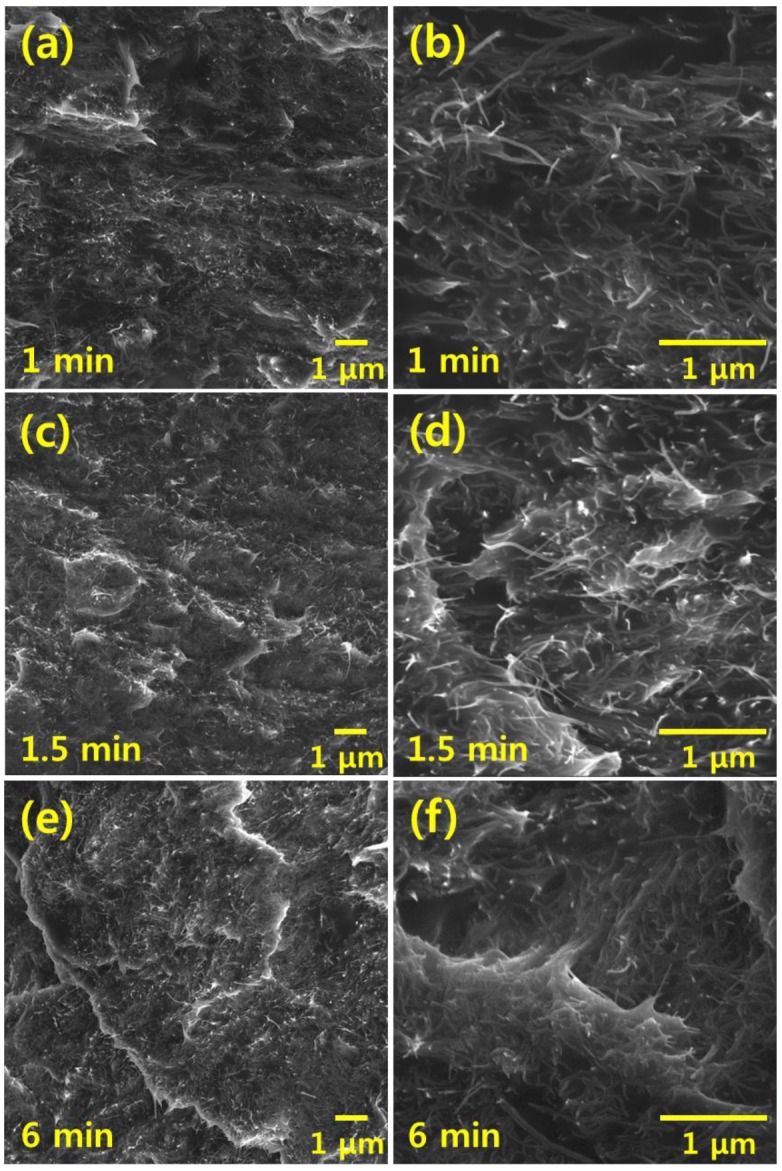
SEM images showing morphology of CNT/PDMS composites fabricated using three-roll milling for (**a**), (**b**) 1 min, (**c**), (**d**) 1.5 min, and (**e**), (**f**) 6 min.

**Figure 6 materials-12-03823-f006:**
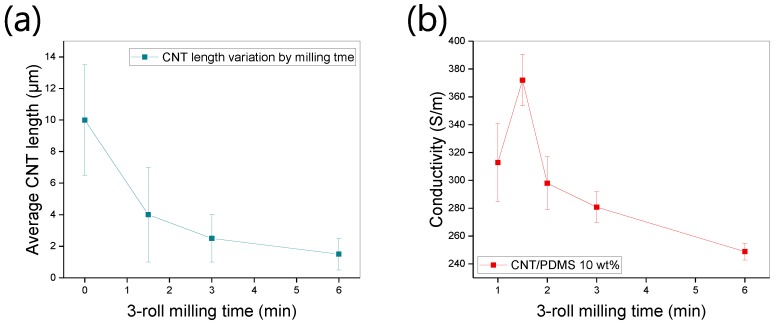
(**a**) Changes in CNT length with increasing three-roll milling time. (**b**) Electrical conductivity of CNT composite films obtained by milling for various times. The increase in electrical conductivity as the milling time increases from 1 to 1.5 min is due to the greater dispersion stability. After 1.5 min of milling time, the conductivity decreases because the decrease in CNT length reduces the number of CNT contacts.

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
