# Peer review of "Effect of Dispersion by Three-Roll Milling on Electrical Properties and Filler Length of Carbon Nanotube Composites"

_materials, 2019, doi:10.3390/ma12233823_

Round 1

Reviewer 1 Report

Please see attached pdf document.

Thank you.

Author Response

We deeply appreciate the comments of the reviewer and have addressed them in the revised manuscript and response letter.

Reviewer 2 Report

The work of Ha et al. described influences of three-roll milling duration on the CNT length and electrical conductivity. The study is carried out on CNT-10wt.% composites, which were milled during different time. The authors investigated effect of homogeneity on electrical conductivity.

The article is classified by correct journal and article type. The references are given in a correct way. The introduction part depicts current state of the art. In the chapter “Materials and Methods” describing of some analysis are missing and should be completed. Results of the study are given in a logical way and are discussed in detail. I would recommend publishing of paper after minor revision.

Line 33: The conductive pathway in the composite improve its electrical properties. Either pathways or improves Section “Materials and Methods”: please describe How large was the gap between rolls Which technique was used for Raman spectra (Fig. 3) and particle size distribution (Fig. 4) Line 81: To disperse the CNTs in the polymer, we performed three-roll milling to apply mechanical shear forces. Please rephrase the sentence: The dispersion of CNTs in polymer was performed by applying of mechanical shear forces in three-roll milling. Figure 6 (a): please rename the y-axis “Average CNT length” Figure 6 (a) and (b): please add in (a) the values for 1 and 10 minutes, than it is for reader easier to follow and to compare the results. Also please add, if possible a value in (b) for conductivity at the time zero. The standard deviations shall appear in red as the values. Lines 145-147: the information is already given in lines 121-124. Please delete. Line 155: With increasing milling time, the dispersion state becomes more stable. What the author means by stable? Homogeneous distributed CNTs or increase of viscosity? Please describe in detail.

Author Response

(The authors gave the same response as above.)

Reviewer 3 Report

This paper shows the experimental results of CNTs/polydimethylsiloxane composites relating to their conductivity of the composites and included CNTs lengths treated by three-roll milling. My comments are listed below.

・I feel this paper needs to show the Raman spectra taken from pristine CNTs and three-rolled milled CNTs, not CNTs composites, to discuss the structural damages for CNTs after milling.

・I think used CNT are forming large aggregation. The aggregation is very important information for filler applications because of these aggregations act as negative effects. Therefore, SEM images of bulk CNT morphologies are also needed to include to estimate the degree of CNT's aggregation relating to treatment time.

・This paper needs to state more experimental details on how to make composite films such as the percentage of CNT in the composites in the experimental section.

Author Response

(The authors gave the same response as above.)

Round 2

Reviewer 3 Report

The revised manuscript has been improved along with the referee's comments. This paper shows enough novelty and importance. Therefore, I recommend the publication of this paper.